# Gut bacterial communities in roadkill animals: A pioneering study of two species in the Amazon region in Ecuador

Manuel Alejandro Coba-Males[1], Magdalena Díaz[1], C. Alfonso Molina[1], Pablo Medrano-Vizcaíno[2,3,4], David Brito-Zapata[4,5], Sarah Martin-Solano[6], Sofía Ocaña-Mayorga[7], Gabriel Alberto Carrillo-Bilbao[1], Wilmer Narváez[1], Jazzmín Arrivillaga-Henríquez[1], Manuela González-Suárez[2], Sandra Enríquez[1]*, Ana Poveda[1]*

1 Grupo de Investigación en Biodiversidad, Zoonosis y Salud Pública (GIBCIZ), Facultad de Ciencias Químicas (FCQ), Facultad de Ingeniería Química (FIQ), Facultad de Medicina Veterinaria y Zootecnia (FMVZ), Instituto de Salud Pública y Zoonosis (CIZ), Universidad Central del Ecuador, Quito, Ecuador, 2 Ecology and Evolutionary Biology, School of Biological Sciences, University of Reading, Reading, United Kingdom, 3 Grupo de Investigación Población y Ambiente, Universidad Regional Amazónica IKIAM, Tena, Ecuador, 4 Red Ecuatoriana para el Monitoreo de Fauna Atropellada – REMFA, Quito, Ecuador, 5 Museo de Zoología & Laboratorio de Zoología Terrestre, Instituto iBIOTROP, Universidad San Francisco de Quito USFQ, Quito, Ecuador, 6 Grupo de Investigación en Sanidad Animal y Humana (GISAH), Carrera Ingeniería en Biotecnología, Departamento de Ciencias de la Vida y la Agricultura, Universidad de las Fuerzas Armadas —ESPE, Sangolquí, Ecuador, 7 Centro de Investigación para la Salud en América Latina, CISeAL Pontificia Universidad Católica del Ecuador, Quito, Ecuador

* ienriquez@uce.edu.ec (SE); apoveda@uce.edu.ec (AP)

**Data Availability Statement:** All relevant data are within the paper and its Supporting information files.

## Abstract

Studying the microbial communities within the gastrointestinal tract of vertebrate species can provide insights into biodiversity, disease ecology, and conservation. Currently, we have very limited understanding of the composition of endogenous microbiota in wildlife, particularly in high biodiversity tropical areas. Knowledge is limited by the logistical and ethical challenges of obtaining samples for free-living animals. Roadkill carcasses offer a largely untapped source for biological material, including endogenous gut microbiota. These animals that have died on roads due to collisions with vehicles are suitable for accessible, opportunistic sampling. Here, we used metabarcoding for the $V_3$—$V_4$ region of the *16S rRNA* gene in gut samples of nine roadkill samples collected from a road in Ecuador representing two vertebrate species: the speckled worm lizard (*Amphisbaena bassleri*) and the smooth-billed ani (*Crotophaga ani*). We successfully identify microbial phyla in both samples including Firmicutes, Bacteroidetes, and Proteobacteria for *A. bassleri*, and Firmicutes and Actinobacteria for *C. ani*. Our study provides the first description of the gut microbiota for these two vertebrates, and demonstrates the feasibility of studying endogenous microbial communities from roadkill material that can be opportunistically collected and preserved in biobanks.

## Introduction

In recent years, studies on the gut microbial communities of wildlife have increased considerably. However, most reports have focused on captive non-human vertebrates from laboratories

**Funding:** Corporación Ecuatoriana para el Desarrollo de la Investigación y Academia - CEDIA (through its CEPRA-XVI-2022 program to AP), as part of the project titled "Estudio de parásitos y microbioma de fauna silvestre en dos de las zonas más biodiversas del planeta: Los Andes Tropicales y Chocó-Darién en Ecuador" fund (https://cedia.edu.ec/servicio/fondo-idi-universidades/cepra-xvi-2022/). Dirección de Investigación de la Universidad Central del Ecuador for the Proyecto Senior 2021 fund under project DI-CONV-2021-16 to WN (https://www.uce.edu.ec/web/di). The University of Reading for Seed Funding from the School of Biological Sciences an International PhD studentship to Pablo Medrano-Vizcaíno (ref GS19-042).

**Competing interests:** The authors have declared that no competing interests exist.

or zoos using fecal samples [1], which likely do not reflect the reality of the microbiome composition in free living organisms [2]. There is a gap of knowledge regarding the endogenous microbial communities of wild animals [3], which is essential to understand natural biodiversity as well as the impact that different natural and anthropogenic factors have on enteric microbial communities.

Ecuador encompasses one of top biodiversity hotspots on the planet, the Tropical Andes [4] and is home to 4,801 vertebrate species (Instituto Nacional de Biodiversidad, http://inabio.biodiversidad.gob.ec/. Accessed on 15 October, 2024). Many of these species die due to collisions with vehicles on Ecuadorian roads, with birds and reptiles being the most frequently roadkill taxa [5]. While wildlife-vehicle collisions are a worrying conservation issue [6], these roadkill specimens offer a sampling opportunity for biological information [7]. Sampling roadkill has no direct negative impact to wildlife [8] and poses few, if any, of the ethical or logistical concerns associated with sampling free-living organisms. Roadkill can offer opportunities to sample poorly studied groups like reptiles [9] and many birds [10], and can contribute to a better understanding of the microbiota associated with native species in Ecuador [11, 12].

Here we showcase the value of samples from roadkill specimens by providing the first initial description of the gut bacterial communities of two vertebrates: the reptile *Amphisbaena bassleri* Vanzolini, 1951 and the bird *Crotophaga ani* Linnaeus, 1758. *A. bassleri* is a blind reptile with fossorial behavior, which feeds on invertebrates [13]. *C. ani* is a bird belonging to Cuculidae family, distributed along South America. These birds feed on insects, small reptiles, and fruits [14]. We used samples from nine specimens collected from the Napo province, Ecuador [15] and next-generation sequencing (NGS) of the hypervariable V$_3$—V$_4$ region from the *16S rRNA* gene to describe for the first time the gut bacterial community of two native Ecuadorian species. We also qualitatively evaluated the relationship between the time elapsed between death and sample preservation and the composition of bacterial communities, as an approach to understand the natural decomposition process.

## Materials and methods

### Sample collection

We used samples from carcasses collected during a systematic roadkill survey of 240 km of roads in the Tropical Andes in the Amazonian province of Napo, Ecuador [15]. Sampling was carried out between September 19[th], 2020, and March 23[rd], 2021; all dead specimens found on the roads were taken to the laboratory for tissue dissection, and samples were preserved in a biobank at -80˚C until their processing, as previously described [7]. Of these, we selected samples for this study that met essential requirements: 1) those belonging to animals from the same species, 2) those in which the roadkill animal (carcass) was relatively intact after the accident as required to allow dissection and obtention of uncontaminated intestine samples. From the 590 specimens collected, only nine specimens from two species met these criteria and could be included in the study. Therefore, the gut samples selected for this work were from four specimens of the reptile *A. bassleri* and five specimens of the bird *C. ani*. Details of the samples used, including timing and location of collection are provided in Table 1.

The permissions used to collect samples were granted by the Ministerio del Ambiente, Agua y Transición Ecológica (MAATE) from Ecuador with No. MAAE-DBI-CM-2021-0215 and MAAE-ARSFC-2020-0791.

### DNA extraction and sequencing

Total genomic DNA (gDNA) was isolated from 15 mg of each gut sample with ZymoBIO-MICS™ DNA Miniprep Kit (Zymo Research, Irvine, CA, United States) according to the

**Table 1. Description of the samples from two vertebrate species analyses for this study.**

| Sample | Species | Estimated time since death | Landscape | Latitude | Longitude |
|--------|---------|---------------------------|-----------|----------|-----------|
| SW001 | *Amphisbaena bassleri* | 0 hours | Altered area | -0.84761 | -77.78706 |
| SW002 | *Amphisbaena bassleri* | 0 hours | Altered area | -0.85528 | -77.79068 |
| SW003 | *Amphisbaena bassleri* | 2 hours | Altered area | -0.77437 | -77.79246 |
| SW004 | *Amphisbaena bassleri* | 6 hours | Unaltered area | -0.82108 | -77.77457 |
| SW005 | *Crotophaga ani* | 1 hours | Unaltered area | -1.10526 | -77.79806 |
| SW006 | *Crotophaga ani* | 1 hours | Altered area | -1.03703 | -77.77536 |
| SW007 | *Crotophaga ani* | 2 hours | Altered area | -1.06761 | -77.63735 |
| SW008 | *Crotophaga ani* | 6 hours | Altered area | -0.39962 | -77.82664 |
| SW009 | *Crotophaga ani* | 48 hours | Unaltered area | -1.04709 | -77.78982 |

Description of the samples including the biobank sample ID, estimated time since death at the time of collection, and the coordinates and surrounding landscape of the collection site described as "Altered area" for landscapes with human land uses (agriculture, pastureland, or built-up areas) and "Unaltered area" for areas with natural land covers.

manufacturer's protocol, quantified with a Qubit® Fluorometer (Invitrogen, Life Technologies, CA, USA) and stored immediately at -20°C for subsequent analysis. To identify bacterial communities, a single amplicon of approximately 550 bp corresponding to the $V_3$—$V_4$ hypervariable region of prokaryotic *16S rRNA* gene was amplified with the forward primer 341F (5'-CCTACGGGNGGCWGCAG-3') and the reverse primer 805R (5'-GACTACHVGGGTATCTAATCC-3') [16]. Each PCR reaction contained 12.5 µL 2x KAPA HiFi HotStart Ready Mix (Kapa Biosystems Inc., MA, USA) which includes 0.3 mM dNTPs, 2.5 mM $MgCl_2$, and 0.5 U of HiFi DNA Polymerase), 0.5µL of each primer (0.2 µM), 12.5 ng of template gDNA, and PCR-grade water to reach a final volume of 25 µL. Thermal cycle conditions according to [17] consisted of an initial denaturation at 95°C for 3 min, followed by 25 cycles at 95°C for 30 sec (denaturation), 55°C for 30 sec (annealing), and 72°C for 30 s (extension). This was followed by a final extension at 72°C for 5 min. The resultant PCR products were purified using Agencourt AMPure XP (Beckman Coulter, Brea, CA, USA). Finally, the preparation of libraries was based on dual-index barcodes of Nextera XT Index Kit (Illumina Inc., San Diego, CA, USA) according to manufacturer's instructions. The libraries were purified, quantified and sequencing on Illumina MiSeq platform with paired-end reads of 300 bp using MiSeq reagent kit v3 (600 cycles PE) (Illumina Inc., San Diego, CA, USA).

## Metabarcoding sequence processing

Initially, quality of the raw read data was checked visually with FastQC v.0.12.1 [18]. The bacterial taxonomic annotation was done with the open-source single software platform mothur v.1.45.0 [19] by following the standard operating procedure to process *16S rRNA* gene sequences from Illumina's MiSeq platform described on the mothur website (https://mothur.org/wiki/miseq_sop/) (accessed on 28 June 2023) [20]. Paired-end reads were merged into contigs and then filtered to retain sequences with a minimum overlap of 20 bp, minimum length of 350 bp and a maximum of 560 bp. Additionally, those sequences with homopolymers longer than 14 bp and/or ambiguous nucleotides were removed. The resulting sequences were deduplicated to reduce redundancy and cluster unique sequences. These unique sequences were aligned against the SILVA release 132 SSU Ref NR dataset [21], which was previously customized for $V_3$—$V_4$ region. This region was defined between positions 6,428 and 23,440 within the non-redundant SILVA v132 reference alignment. Sequences that did not cover the full

alignment were filtered starting at position 2 and ending at position 17,012. Subsequently, the alignments underwent processing to remove columns containing gap or dot characters. Once again, we performed a new round of sequence deduplication in case the previous step had generated additional redundant sequences. To denoise data, sequences that were very similar (up to one nucleotide difference between them) were preclustered, and then using the VSEARCH algorithm [22], chimeras were removed from all samples when an abundant sequence was flagged as chimeric in one sample. The taxonomic annotation of the sequences was based on reference taxonomy database of SILVA release 132 and done by the Wang method [23] which use a naïve Bayesian classifier with a 70% bootstrap threshold. Sequences that belong to taxa like chloroplasts, mitochondria, and Eukaryota were unsuitable for our purpose, and thus, were removed from the dataset. The final sequences were clustered by the OptiClust algorithm [24] into operational taxonomic units (OTUs) with the similarity cut-off value of 99%. The most abundant sequence of each OTU was picked to do a consensus taxonomy classification from phylum to genus level and determinate the number of representative sequences.

## Microbial communities analysis

Microbial community data were analyzed using R version 4.3.1 [25] and RStudio version 2023.6.0.421 [26]. Final output files from mothur's pipeline were imported to R with *phyloseq* package version 1.44.0 [27] to generate a phyloseq object with added metadata information for each sample. OTUs were grouped at genus level, and we reviewed the kingdom and phylum levels to remove archaea, unknown taxa, unclassified bacteria and all singletons. Samples were separated by vertebrate species (*Amphisbaena bassleri* and *Crotophaga ani*) into two phyloseq objects for further independent analysis. The composition of bacterial communities was explored through bar plots of relative abundance to identify the most abundant phyla. We established a threshold value to retain phyla with a relative abundance higher than 3% in at least one sample. Those phyla with a relative abundance below this threshold were grouped in an "Others" category.

## Diversity analyses

**Alpha diversity.** We adapted the methods from [11] with α-diversity indexes Chao1, Shannon and Simpson calculated with the *phyloseq* package in R using the function estimate_richness. Downstream analyses of both vertebrate species were done using OTUs abundances belonging to the Genus-level taxonomic classifications. To compare median similarities in alpha diversity of genus data among samples from each species, we performed a Kruskal-Wallis test followed by a post hoc Bonferroni test using the free software for scientific data analysis, PAST [28]. We used adjusted *p*-values, considering significant results at $p < 0.05$.

Rarefaction curves for gut microbiota in *A. bassleri*, and *C. ani* samples were estimated by the *ranacapa* package version 0.1.0 [29] in R, using steps of 600 samples with the back-end functions. The overall absolute abundance at family level was visualized on heatmaps generated with *ComplexHeatmap* package version 2.16.0 [30, 31] in R. To select the 25 most abundant families, we did a logarithmic transformation to avoid overplotting, for *A. bassleri* sequences, the cut-off was $\log(x + 1) > 10.3$ and for *C. ani* sequences, the cut-off was $\log(x + 1) > 17.9$. The selected families were hierarchically clustered with the unweighted pair group method (UPGMA on Euclidean distances). Finally, we used a Venn diagram to represent the core microbiome in *A. bassleri* and *C. ani* with *VennDiagram* package version 1.7.3 [32]. For this purpose, we selected genus level with cut-off values of 90% of prevalence and 0.01% of abundance [11].

**Beta diversity.** To assess β-diversity we used a principal coordinate analysis (PCoA). Prior to this analysis, sampling depth was standardized by incorporating the minimum sequencing depth value across all samples for each specimen. Specifically, the sequencing depths were 26,185 for *A. bassleri* and 44,409 for *C. ani* samples. A distance matrix using the Bray-Curtis distance measure was used to quantify the differences between samples. Then, the PCoA was applied using the ordinate function from *phyloseq* package to represent the variability between samples and evaluate their relative arrangement in a multidimensional space [33].

## Results

### Sequencing data analysis

In this study, next-generation sequencing results were employed to describe the gut microbiota of 4 *A. bassleri* and 5 *C. ani* from roadkill specimens. A total of 2,201,434 paired-end raw sequences were generated from the nine gut samples, representing 576.2 Mb of data. For *A. bassleri*, the reads ranged from 69,538 to 268,832, with a mean and SD of 183,542 ± 90,721 per sample. For *C. ani*, the number of reads ranged from 121,826 to 518,250, with mean and SD of 293,453 ± 147,704 per sample. All reads were assembled in 1,100,717 contigs with an average read length of 431 bp. The quality control filter implemented by mothur eliminated 31.6% of the sequences. The filtered assembled sequences included 752,856 contigs, most of which had lengths ranging between 439 and 515 bp. Clustering sequences to reduce redundancy resulted in 309,872 unique sequences that were aligned with the customized $V_3$-$V_4$ region of the non-redundant SILVA v132 reference dataset. After the alignment refinement, the number of sequences was reduced to 703,791. After data denoising, pre-clustering, and removal of chimeras and undesirable lineages, the final number of sequences was 695,970. These represented 38,777 OTUs which were classified into six taxonomic ranks (Kingdom, Phylum, Class, Order, Family, Genus) and then binned into 313 and 416 genera for *A. bassleri*, and *C. ani*, respectively, of which 295 and 381 were unique to each vertebrate species (Table 2).

NCBI's Sequence Read Archive (SRA) with the BioProject is available by the accession number: PRJNA1061813.

### Microbial communities analysis

**Amphisbaena bassleri.** We identified 19 unique phyla associated to *A. bassleri* specimens (Fig 1A). The complete data obtained in the microbiome analyses of the four specimens of *A. bassleri* collected can be consulted in S1 Data. Firmicutes was a superabundant phylum common to all samples. Both samples collected immediately after collision (estimated time since death 0 hours, SW001 and SW002) showed a high relative abundance of Firmicutes, 95.42% and 98.13%, respectively. Sample SW003 (estimated time since death 2 hours) had many

**Table 2. Number of unique taxa.**

|  | *Amphisbaena bassleri* | *Crotophaga ani* |
|---|---|---|
| Phylum | 19 | 21 |
| Class | 35 | 43 |
| Order | 83 | 99 |
| Family | 136 | 170 |
| Genus | 295 | 381 |

Description of unique taxa obtained in the gut samples from four reptile species of *A. bassleri* and five bird species of *C. ani*.

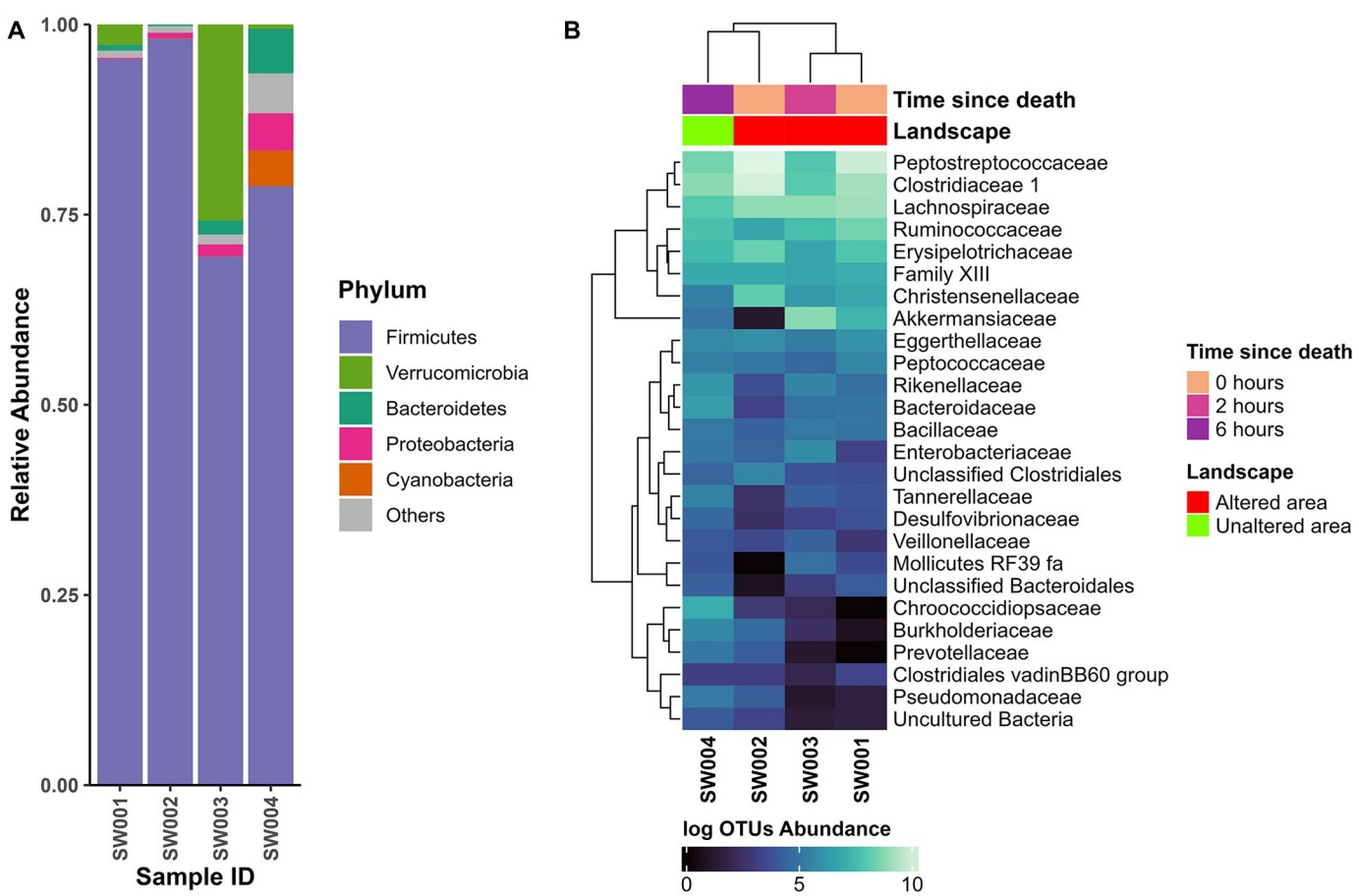

**Fig 1. Microbiota composition in roadkill samples of *A. bassleri*.** (**A**) Stacked bar plots of relative abundance labelling the most abundant phyla (>3%), and the category "Others" grouping all phyla with relative abundance <3%. (**B**) Heatmap at family level depicting the most abundant 25 different OTUs according to postmortem hours and hierarchical clustering of the OTUs and the samples.

Verrucomicrobia, comprising 25.77% of the sequenced material. In contrast, this phylum was rare (2.69%) in sample SW001 which was collected near shrubs close to a human settlement. Bacteroidetes and Proteobacteria appeared to become more abundance with longer times since death, both found as <1% in the two samples collected at 0 hours since death, <2% in the sample collected after 2 hours, and to 5.87% and 4.89% respectively in the sample collected after 6 hours (sample SW004).

We identified 33 families within Proteobacteria, 29 within Firmicutes, 23 within Actinobacteria, and 11 within Bacteroidetes. The heatmap displays the distribution of the 25 most abundant families (Fig 1B). Both samples SW001 and SW002 (0 hours postmortem) showed similar abundances in families such as Peptostreptococcaceae, Clostridiaceae_1, Lachnospiraceae, Erysipelotrichaceae, and Family_XIII. Of these, Peptostreptococcaceae, Clostridiaceae_1, and Lachnospiraceae were the most abundant in both samples. Sample SW003 was primarily dominated (> 25% relative abundance) by Akkermansiaceae and Lachnospiraceae families, although other families such as Clostridiaceae_1, Ruminococcaceae, and Peptostreptococcaceae exhibited a relative abundance close to 10%. In the sample SW004 (6 hours postmortem), we observed two predominant families within its bacterial community: Clostridiaceae_1 and Peptostreptococcaceae. Only three families were not present in some samples, Chroococcidiopsaceae and

Prevotellaceae were missing from sample SW001, while Mollicutes_RF39_fa was not present in sample SW002. Additionally, approximately 19% of the total composition comprised various other families. Although some changes in communities appear in samples collected at different estimated times since death, the limited sample size precluded further analyses.

**Crotophaga ani.** Gut microbial composition in these five bird specimens revealed 21 unique phyla. The complete data obtained in the microbiome analyses of the *C. ani* specimens can be consulted in S2 Data. For all samples, Firmicutes and Actinobacteria were the most common abundant phyla (Fig 2A). The sample SW006 found in road surrounded by forest and pastureland was mainly constituted by Firmicutes in 97.90%.

Also, Proteobacteria were present in all the samples, with sample SW007 showing the highest proportion (16.53%) (Fig 2A). Epsilonbacteraeota was detected in two samples, SW005 and SW007, the latter exhibiting the highest relative abundance at 18.83%. The sample SW005, collected from forest shrubs at 1 hour postmortem, had the highest relative abundance of Actinobacteria (25.34%), and was the only sample that showed Chlamydiae (5.02% of its composition). Actinobacteria was also detected in samples SW007, SW008, and SW009 with relative abundances of 12.80%, 17.52%, and 20.08%, respectively. Despite the relative abundance of Bacteroidetes being <1% in most of the samples, sample SW009 exhibited a relative abundance of 3.27% for this phylum.

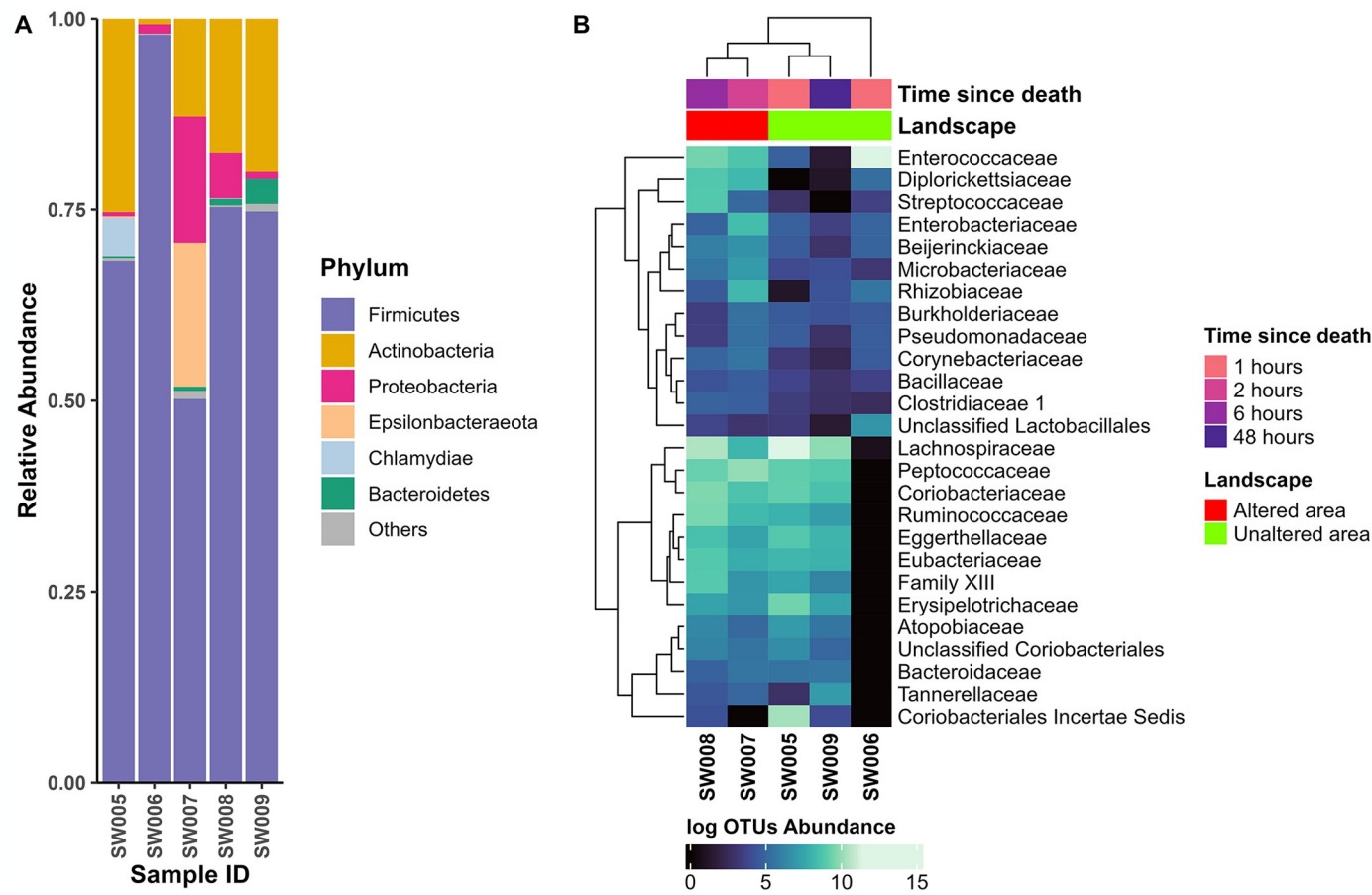

**Fig 2. Microbiota composition in roadkill samples of *C. ani*.** (**A**) Stacked bar plots of relative abundance labelling the most abundant phyla (>3%), and the category "Others" grouping all phyla with relative abundance <3%. (**B**) Heatmap at family level depicting the most abundant 25 different OTUs according to postmortem hours and hierarchical clustering of the OTUs and the samples.

We identified 41 families in Proteobacteria, 38 in Actinobacteria, 30 in Firmicutes, and 12 in Bacteroidetes. In *C. ani* we found no clear differences in communities associated to time of death or the landscape where samples were found (Fig 2B). For example, the two samples collected at 1 hour postmortem in unaltered landscapes (SW005, SW006) showed very different patterns (Fig 2B). The abundance distribution of the top 25 bacterial families showed that eleven families, including Atopobiaceae, Bacteroidaceae, Coriobacteriaceae, unclassified Coriobacteriales, Eggerthellaceae, Erysipelotrichaceae, Eubacteriaceae, Family_XIII, Peptococcaceae, Ruminococcaceae, and Tannerellaceae, were entirely absent in sample SW006 (Fig 2B). This observation could be related with low diversity of this sample (see alpha diversity). Moreover, the family Coriobacteriales Incertae Sedis was not present in samples SW006 and SW007.

## Alpha diversity

In both species sample size was limited, and results should be interpreted with caution. To understand the diversity of the samples, we calculated Shannon, Simpson and Chao-1 indexes (Table 3). For *A. bassleri* samples excluding sample SW003, the observed OTUs were closer to the estimated number of OTUs by Chao1 index (Table 3), therefore we reported at least more than 87% of all communities present in each sample. We found significant differences in bacterial diversity across samples (Kruskal-Wallis test H = 20.35, $p$ = 0.00013) with SW003 being different to SW002 ($p$ = 0.0106) and SW004 ($p$ = 0.00032). Differences based on pairwise comparison using the Bonferroni test). According to Shannon's and Simpson's indexes, SW004 was the sample with higher diversity. Bacterial diversity did not consistently change with time since death.

For *C. ani*, sample SW006 had the lowest diversity according to Shannon's and Simpson's indexes, while the other samples had moderate diversity (Table 3). Analysis revealed that in SW006 we identified 68% of the estimated OTUs by Chao1 index (Table 3), while other samples reached more than 87% of estimated OTUs. Bacterial diversity in *C. ani* showed significative differences between samples (H = 16.52, $p$ = 0.0023), with sample SW007 different from SW005 ($p$ = 0.013) and SW008 ($p$ = 0.0045). Despite of the existence of differences in bacterial richness, variation was not clearly associated with the time since death.

## Beta diversity

There was no clear pattern of microbiome composition related with the time of death based on Principal Coordinate Analysis (PCoA). For *A. bassleri* samples SW001 and SW002 at 0 hours since death were similar, but samples collected at 2 hours (SW003) and 6 hours (SW004) were uniquely distributed (Fig 3A). For *C. ani* (Fig 3B) both samples collected at 1 hour since death

**Table 3. Alpha diversity indexes for samples collected of *A. bassleri* and *C. ani*.**

| SampleID | *Amphisbaena bassleri* | | | | *Crotophaga ani* | | | | |
|---|---|---|---|---|---|---|---|---|---|
| | SW001 | SW002 | SW003 | SW004 | SW005 | SW006 | SW007 | SW008 | SW009 |
| OTUs Observed | 156 | 157 | 170 | 156 | 199 | 82 | 209 | 218 | 133 |
| Shannon | 2.5077 | 2.1036 | 2.7228 | 3.2328 | 2.3375 | 0.3936 | 2.9774 | 2.7680 | 2.5425 |
| Simpson (1-D) | 0.1702 | 0.2136 | 0.1344 | 0.0972 | 0.1679 | 0.8766 | 0.1182 | 0.0875 | 0.1305 |
| Chao1 | 179.25 | 171.62 | 215.04 | 165.50 | 215.71 | 120.25 | 223.62 | 224.84 | 152.00 |
| T. s. death | 0 hours | 0 hours | 2 hours | 6 hours | 1 hours | 1 hours | 2 hours | 6 hours | 48 hours |

T. s. death, Time since death.

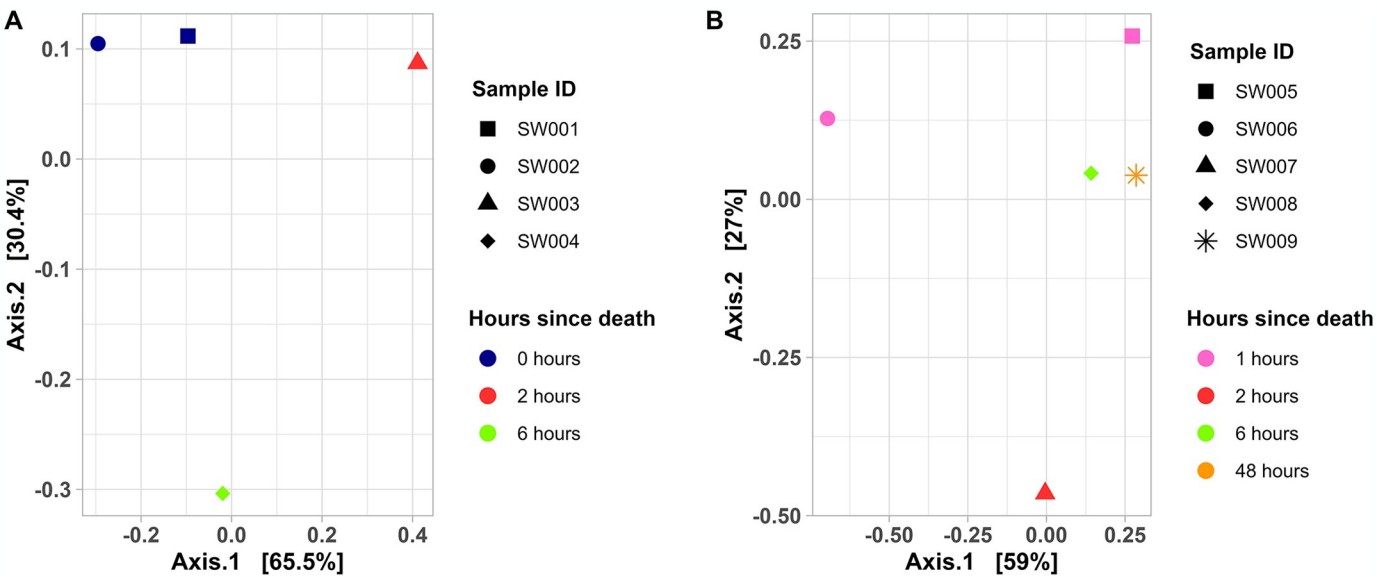

**Fig 3. Principal Coordinates Analysis of bacterial communities among (A)** *A. bassleri*, **and (B)** *C. ani* **samples.** Hours since death indicated the estimated time since death due to wildlife-vehicle collision of the sampled roadkill specimens.

(SW005 and SW006) differ, but samples SW008 and SW009 were near each other, despite their difference in collection time (6 and 48 hours since death).

## Core gut microbiota

Analyzing the common bacterial taxa present in the gut microbiota among samples from each animal species allowed us to identify certain genera that could represent the core microbiota for *A. bassleri* and *C. ani*.

The core gut microbiota structure of *A. bassleri* included 41 genera shared among all four specimens (Fig 4A). Most of these genera belonged to the order Clostridiales (phylum Firmicutes), which represented 78.05% of the core microbiota and included 9 families: Christensenellaceae, Clostridiaceae_1, unclassified Clostridiales, Clostridiales_vadinBB60_group, Family_XIII, Lachnospiraceae, Peptococcaceae, Peptostreptococcaceae, and Ruminococcaceae. At the phylum level, Firmicutes was the major contributor in the core microbiota, with Bacteroidetes and Actinobacteria also present. Additionally, coliform bacteria that inhabit the intestines of most animal species and belong to the family Enterobacteriaceae (phylum Proteobacteria) were also part of the core microbiota for *A. bassleri*.

The core microbiota for *C. ani* was limited just to three genera found in all samples: *Methylobacterium* (family Beijerinckiaceae), *Bacillus* (family Bacillaceae), and *Pseudomonas* (family Pseudomonadaceae), with most genera commonly found in single individual samples (Fig 4B).

## Discussion

The present study represents a pioneering effort aimed at describing the endogenous microbial communities of free-living vertebrates in the Amazon region of Ecuador. Our results provide the first description of the bacterial community profiles in two vertebrate species, contributing to a topic that is poorly understood particularly in this high biodiverse region. The two studied

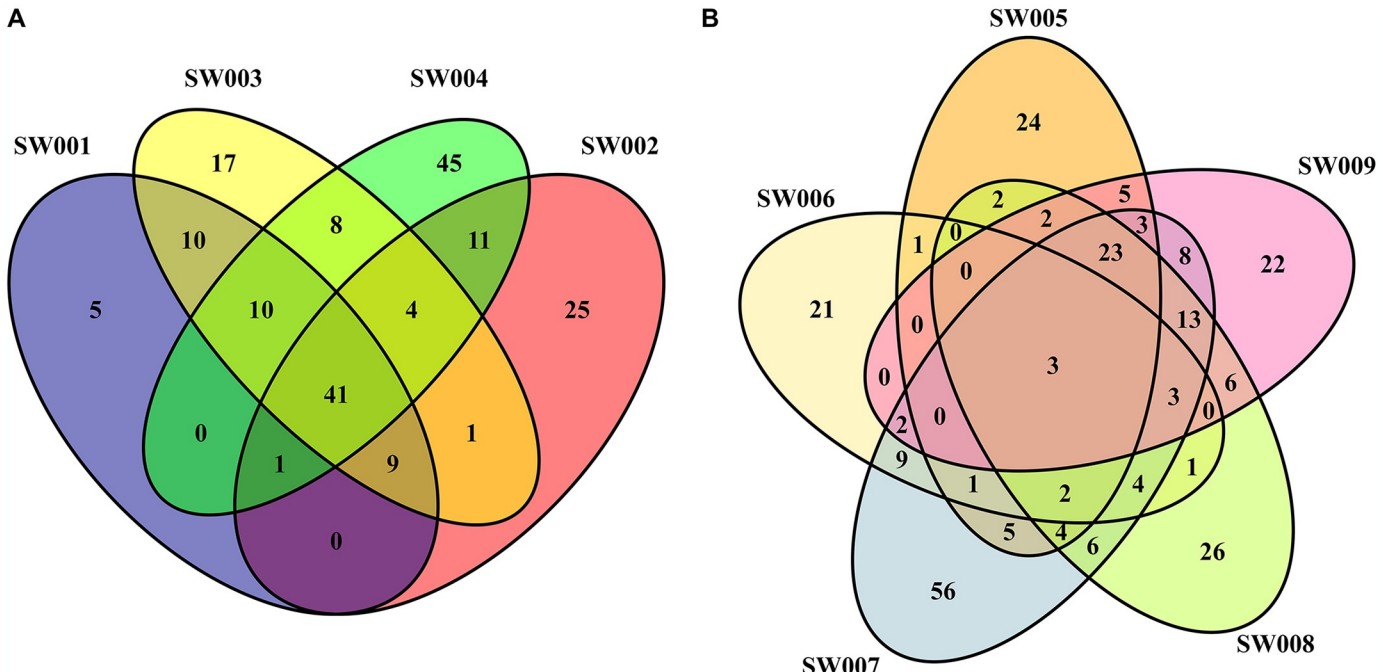

**Fig 4. Gut core microbiota.** Gut core microbiota at Genus level showing the number of shared OTUs across all the samples for (**A**) *A. bassleri* and (**B**) *C. ani*.

vertebrates were found as roadkill in our study area but were not the only species detected [5] which highlights the opportunities to sample and describe microbiomes in wildlife using this approach. We hope our study motivates further explorations but acknowledge limitations with our sample size. Roadkill surveys primarily aim to quantify road impacts but offer an additional opportunity to sample roadkill animals. These samples general represent diverse taxonomic group and can include threatened species from which live capture may not be possible or ethical, but can be limited by relatively few replicates or samples per taxon. Previous studies using other types of opportunistic sampling, and thus having limited sample sizes, have contributed to the literature, for example collecting data from stranded cetaceans [34, 35] or dolphins [36, 37]. Another challenge of using dead specimens like roadkill is that during the postmortem period there could be an uncontrolled proliferation of microorganisms leading to changes in bacterial community profiles because of microbial colonization on the surfaces of animal carcasses [38]. This aspect should be further explored but our preliminary analyses did not find strong clear pattern of changes associated to the duration of the postmortem period.

## Gut microbiota in *Amphisbaena bassleri*

All samples of *A. bassleri* were primarily dominated by Firmicutes, which ranged from 69.51% to 98.13%, while Bacteroidetes (0.22–5.86%), Proteobacteria (0.17–4.89%), and Verrucomicrobia (0.56–25.77%) constituted relatively minor components of the gut microbiota. No other studies have reported on the gut microbiome composition in the family Amphisbaenidae or any families within the infraorder Amphisbaenia. There are some microbiome analyses of species of the family Lacertidae, which is the most phylogenetically related to the genus *Amphisbaena*. However, these reports were not obtained from direct gut tissue samples. For the Lacertidae lizard *Eremias argus*, the predominant phyla are Firmicutes, Proteobacteria,

Actinobacteria, Bacteroidetes and Verrucomicrobia [39]. The first four phyla also showed high abundance in lizard species of the genus *Podarcis* [40]. In addition, the composition is similar to the endogenous microbiota reported for other reptiles, such as the wild Burmese pythons (*Python bivittatus*), Gopher tortoises, Australian saltwater crocodiles (*Crocodylus porosus*), American alligators (*Alligator mississippiensis*), cottonmouth snakes (*Agkistrodon piscivorus*), and some lizards (*Phrynocephalus vlangalii*), where Firmicutes and Bacteroidetes consistently were identified as the dominant phyla [3, 41]. The abundance of these two phyla has been associated with metabolic processes during fasting periods or active digestion, respectively [3]. Additionally, Proteobacteria has been identified as a common phylum in reptile microbiomes, showing variable abundance in samples obtained from the small gut or cloaca [42], while Verrucomicrobia can reach up to 4% in some reptilian species [43], and in other cases, even be the dominant phylum [44].

We found variability in gut bacterial communities of *A. bassleri*, which was not clearly associated to differences in estimated time since death. These variations may be associated with environmental conditions, such as the surrounding landscape, as the gut microbiome in reptiles begins to develop from environmental exposure during the juvenile stages of life [3]. Furthermore, temperature may play a significant role in the variability and composition of the gut microbiome. Samples were collected in a tropical environment with average temperatures ranging from 4.63 to 23.7˚C and relatively high humidity. It is possible that the low temperature and the short interval of time that has elapsed since the death in the animal did not allow optimal growth in terms of abundance for mesophilic bacteria that typically colonize the corpse during decomposition.

The PCoA analysis suggested similarity in both fresh samples (collected at 0 hours postmortem) compared to the others. This could reflect similarities in the microbiome of living *A. bassleri*, but further analysis with a larger sample size is necessary. Despite differences, all samples generally had a high predominance of Firmicutes, accounting for 85.45% until 6 hours after death. This finding contrasts with studies in mammalian corpses in which bacterial communities were dominated by Proteobacteria in the early stages, between 0- and 120- hours postmortem, then strongly shifting towards Firmicutes as decomposition progressed [45].

## Gut microbiota in *Crotophaga ani*

The crucial ecological role played by birds has prompted investigations into the composition of their gut microbiome, encompassing both captive and wild species, with efforts to address methodological limitations [10]. While some studies have identified Bacteroidetes and Proteobacteria as the predominant phyla in birds, with low proportions of Firmicutes in wild species [46], our analysis of five *C. ani* specimens primarily revealed a dominance of Firmicutes (50.23–97.90%) and Actinobacteria (0.7–25.34%), with a minimal contribution of Proteobacteria (0.54–16.53%). This composition partly resembles findings in a long-distance migratory swallow (*Hirundo rustica*), where Proteobacteria, Firmicutes, and Actinobacteria were the dominant bacteria [47]. The gut microbiota described in *Clamator glandarius*, a species from the family Cuculidae and phylogenetically related to the genus *Crotophaga*, revealed a higher abundance of Firmicutes and Bacteroidetes [48]. In contrast, *C. ani* showed small relative abundances of Bacteroidetes, ranging from 0.23% to 3.27%. Thus, studies indicate that the microbiome of *C. ani* is predominantly characterized by Firmicutes, with a smaller composition of phyla such as Actinobacteria, Bacteroidetes, and Proteobacteria [49].

The only known microbiome study for the genus *Crotophaga* had a social approach and focused on characterizing the preen gland and body feather microbiota of *C. ani* [50]. Although direct comparisons are not possible, we found one interesting similarity. The preen

gland and feather microbiota of *C. ani* showed bacterial taxa corresponding to the families Pseudomonadaceae and Lachnospiraceae, as well as the genus *Methylobacterium* [50], which are the same as those reported by us in the gut samples of this species. Our results in *C. ani* could be useful in contributing to knowledge in different fields related to this bird.

We also found disparity in the intestinal microbiota among samples, which may reflect changes in time since death, environmental conditions and physiological needs or state [10]. In our limited sample, changes were not clearly associated with time since death. For example, both samples obtained at the same time after death (1 hour), exhibited distinct microbial compositions and did not cluster together in the PCoA analysis. Future work using larger sample sizes would be necessary to establish how microbiomes change with time postmortem in this and other wildlife species. A single sample (collected 2 hours post-mortem) had the phylum Epsilonbacteraeota, with *Helicobacter* as the predominant genus. Some species of *Helicobacter* are known pathogens, and future work would be useful to quantify its prevalence in vertebrates in this region.

Our study offers a first report of endogenous gut microbiota for two wild vertebrate and showcases the potential to use roadkill samples as an innovative source of biological material to characterize bacterial communities in wildlife without causing harm to the animals. While inferences about the effect of time since death or habitat on microbiome diversity could not fully explored due to the small sample size, roadkill samples can offer valuable information about the gut microbiota of live individuals [51].

Care is needed to collect largely intact specimens later carefully dissected in a laboratory, internal sampling (e.g. gut samples) can also help identify microorganisms more likely to be associated to the living hosts. Roadkill samples can offer a valuable and easy to collect (accessible from roads) source of biological material, their collection and preservation in biobanks can facilitate future studies, allow monitoring and detection of changes over time and allow to explore many new questions about the microbial gut communities of many free-living species.

## Supporting information

**S1 Data. OTUs in *A. bassleri* samples.** Complete data obtained in the microbiome analyses of the four specimens of *A. bassleri* collected.
(XLSX)

**S2 Data. OTUs in *C. ani* samples.** Complete data obtained in the microbiome analyses of the five specimens of *C. ani* collected.
(XLSX)

**S1 Table. Sequencing data from two vertebrate species analyses for this study.**
(DOCX)

**S2 Table. Relative abundance at the phylum level in *A. bassleri* gut samples.**
(DOCX)

**S3 Table. The 25 most abundant families in *A. bassleri* gut samples.**
(DOCX)

**S4 Table. Relative abundance at the phylum level in *C. ani* gut samples.**
(DOCX)

**S5 Table. The 25 most abundant families in *C. ani* gut samples.**
(DOCX)

**S6 Table. Shared OTUs among the *A. bassleri* samples.**
(DOCX)

**S7 Table. Shared OTUs among the *C. ani* samples.**
(DOCX)

## Acknowledgments

We thank to the Corporación Ecuatoriana para el Desarrollo de la Investigación y Academia—CEDIA for providing us access to the cluster HPC CEDIA, which was essential for conducting the bioinformatics analyses in the context of this study.

## Author Contributions

**Conceptualization:** C. Alfonso Molina, Sandra Enríquez, Ana Poveda.

**Data curation:** Manuel Alejandro Coba-Males, Magdalena Díaz.

**Formal analysis:** Manuel Alejandro Coba-Males, Magdalena Díaz.

**Funding acquisition:** Wilmer Narváez, Manuela González-Suárez, Sandra Enríquez, Ana Poveda.

**Investigation:** Manuel Alejandro Coba-Males, Pablo Medrano-Vizcaíno, David Brito-Zapata.

**Methodology:** Manuel Alejandro Coba-Males, Magdalena Díaz, Ana Poveda.

**Project administration:** Wilmer Narváez, Ana Poveda.

**Supervision:** C. Alfonso Molina, Sandra Enríquez, Ana Poveda.

**Writing – original draft:** Manuel Alejandro Coba-Males, Ana Poveda.

**Writing – review & editing:** Pablo Medrano-Vizcaíno, David Brito-Zapata, Sarah Martin-Solano, Sofía Ocaña-Mayorga, Gabriel Alberto Carrillo-Bilbao, Wilmer Narváez, Jazzmín Arrivillaga-Henríquez, Manuela González-Suárez, Sandra Enríquez.

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
