## [Decision Letter · Decision Letter 0]

8 Sep 2024

PONE-D-24-20575Gut Bacterial Communities in Roadkill Animals: A Pioneering Study of Two Species in the Amazon region in EcuadorPLOS ONE

Dear Dr. Poveda,

Thank you for submitting your manuscript to PLOS ONE. After careful consideration, we feel that it has merit but does not fully meet PLOS ONE’s publication criteria as it currently stands. Therefore, we invite you to submit a revised version of the manuscript that addresses the points raised during the review process.

We look forward to receiving your revised manuscript.

Kind regards,

Sayed Haidar Abbas Raza

Academic Editor

PLOS ONE

“On behalf of all authors, I disclose any competing interests.”

Reviewers' comments:

Reviewer's Responses to Questions

**Comments to the Author**

1. Is the manuscript technically sound, and do the data support the conclusions?

Reviewer #1: Partly

Reviewer #2: Yes

2. Has the statistical analysis been performed appropriately and rigorously? 

Reviewer #1: No

Reviewer #2: Yes

3. Have the authors made all data underlying the findings in their manuscript fully available?

Reviewer #1: Yes

Reviewer #2: Yes

4. Is the manuscript presented in an intelligible fashion and written in standard English?

Reviewer #1: Yes

Reviewer #2: Yes

5. Review Comments to the Author

Reviewer #1: The authors of this paper have proposed an innovative method the sample the microbiome of the wildlife which are killed in the accidents, this contributes to the ethical and logistical considerations of the sample collection from live animals.

The authors highlight the gut microbiome of the two species which was not studied earlier.

The researchers have an edge over other works die to the research being conducted in the region with a rich biodiversity (Ecuador) which contributes to the knowledge of local wildlife.

Problems

Although the method is good, but the sample size is too small to eliminate any biases this limits the works ability to generalize the findings.

The microbiome also needs to be compared with the live animal's sample. The only roadkill animal research makes data skewed.

The hypothesis needs to be clear, not just description of the microbiome.

The methods and result need to be elaborated more.

Recommendation

Increase sample size of the species.

Include the comparative study of the related species.

Clearly mention the research qu3estin and hypothesis.

Describe the barcoding methods more.

Mention the limitations of the study.

Reviewer #2: The manuscript "Gut Bacterial Communities in Roadkill Animals: A Pioneering Study of Two Species in the Amazon region in Ecuador" reports the gut microbes of Amphisbaena bassleri and Crotophaga ani. These two are wild animal species. Amphisbaena bassleri is a lizard whereas Crotophaga ani is a bird. There are no previous reports on the gut microbiome of these species as it is very difficult to collect samples from them. The authors have used the roadkill carcass of these animals to collect the samples. This is a very good strategy as the samples are collected without harming the wildlife. The authors performed NGS on the hypervariable V3-V4 region from the 16S rRNA. NGS of the hypervariable V3-V4 region from the 16S rRNA gene is a common method for studying bacterial diversity in microbial communities. The authors identified microbial phyla in both samples including Firmicutes, Bacteroidetes, and Proteobacteria for A. bassleri, and Firmicutes and Actinobacteria for C. ani. They also discuss the identified gut microbiota from Amphisbaena bassleri in light of the known literature from other reptiles. Also got microbiota from Crotophaga ani was discussed in reference to findings from other birds. Overall the authors clearly discuss their findings along with limitations of the study.

6. PLOS authors have the option to publish the peer review history of their article (what does this mean?). If published, this will include your full peer review and any attached files.

Reviewer #1: **Yes: **Sabyasachi Mohanty

Reviewer #2: No

---

## [Author Response · Author response to Decision Letter 0]

17 Oct 2024

PONE-D-24-20575

Gut Bacterial Communities in Roadkill Animals: A Pioneering Study of Two Species in the Amazon region in Ecuador

Responses to Editor and Reviewers comments 

Editor:

Response: We have checked all of the style requirements. If we have missed anything, please let us know.

“On behalf of all authors, I disclose any competing interests.”

Response: We agree with that.

Response: We agree.

We require you to either (1) present written permission from the copyright holder to publish these figures specifically under the CC BY 4.0 license, or (2) remove the figures from your submission.

We decided to remove Figure 1.

ANSWER TO THE REVIEWERS:

Reviewer #1: The authors of this paper have proposed an innovative method the sample the microbiome of the wildlife which are killed in the accidents, this contributes to the ethical and logistical considerations of the sample collection from live animals.

The authors highlight the gut microbiome of the two species which was not studied earlier.

The researchers have an edge over other works die to the research being conducted in the region with a rich biodiversity (Ecuador) which contributes to the knowledge of local wildlife.

Problems

Although the method is good, but the sample size is too small to eliminate any biases this limits the works ability to generalize the findings.

The microbiome also needs to be compared with the live animal's sample. The only roadkill animal research makes data skewed.

The hypothesis needs to be clear, not just description of the microbiome.

The methods and result need to be elaborated more.

Recommendation

Increase sample size of the species.

Include the comparative study of the related species.

Clearly mention the research qu3estin and hypothesis.

Describe the barcoding methods more.

Mention the limitations of the study

Answer to Reviewer #1

We thank Reviewer #1 for the comments provided, which are very helpful to improve our work. We will try to answer and argue the questions precisely.

Firstly, we think it is important to explain how the sampling was achieved (previously described [1]. In this study we worked with samples obtained from a previous dedicated and time-consuming field survey that was carried out in a specific period (September 2020-March 2021) involving approximately 5-6 surveys per week, each survey requiring approximately 8 hours per day, by a team of two people (approximately 2000 hours invested). All the roadkill animals in the survey area, a specific geographical location in Ecuador, were collected, resulting in a collection of 590 specimens. Of these, we selected samples for this study that met essential requirements: 1) those belonging to animals from the same species, 2) those in which the roadkill animal (carcass) was relatively intact after the accident as required to allow dissection and obtention of uncontaminated intestine samples. From the 590 specimens collected, only nine specimens met these criteria and could be included in the study. This limitation reflects the difficulty of the fieldwork and collecting non-invasive samples from wildlife (e.g. requiring no manipulation or disturbance to live animals). These samples are described in table 1 of the manuscript.

Table 1. Description of the samples from two vertebrate species analyses for this study. 

.

Sample Species Estimated time since death Landscape Latitude Longitude

SW001 Amphisbaena bassleri 0 hours Altered area -0.84761 -77.78706

SW002 Amphisbaena bassleri 0 hours Altered area -0.85528 -77.79068

SW003 Amphisbaena bassleri 2 hours Altered area -0.77437 -77.79246

SW004 Amphisbaena bassleri 6 hours Unaltered area -0.82108 -77.77457

SW005 Crotophaga ani 1 hours Unaltered area -1.10526 -77.79806

SW006 Crotophaga ani 1 hours Altered area -1.03703 -77.77536

SW007 Crotophaga ani 2 hours Altered area -1.06761 -77.63735

SW008 Crotophaga ani 6 hours Altered area -0.39962 -77.82664

SW009 Crotophaga ani 48 hours Unaltered area -1.04709 -77.78982

The reviewer recommends increasing the sample size. We agree the sample size is small and acknowledge this in the manuscript (lines 238, 267, and 322, applying tracking changes). However, we would argue that novel techniques or findings can be first reported or published based on small samples (eg first report of ingested microplastic on a dolphin based on 3 samples[2], 8 samples used in a study of stable carbon isotope diagnostics published in Plos One[3]). As explained above, we do not have additional samples we can add from our existing specimen collection. To add more samples, we could use two approaches, both of which we argue are not feasible. First, we could collect new roadkill specimens. Beyond the considerable effort this would represent, new samples, if secured, would represent a different time and conditions. It is already known that there are strong microbiome variations between individuals in the same species, even in different intestinal regions of the same individual, even more so if individuals are collected in different geographical areas, and at different times, since factors such as climate, diet or disturbed habitats can have a high impact on microbiomes [4,5]. Moreover, a new field survey does not guarantee we will secure samples (be able to find roadkill animals) for these two species in the necessary conditions. A second option to secure samples, would be to conduct field survey to locate live animals from these species that we would need to handle to collect fecal material, a highly invasive manipulation. In addition to presenting the same spatio-temporal limitations mentioned above, it is precisely a key proposal of our methods that they do not require handling or killing live wild animals that requires permits and brings substantial ethical and welfare concerns. It is worth mentioning finding live species would be particularly challenging for A. basleri, a cryptic species that is difficult to observe in nature. Cryptic species are those that are hard to detect due to their elusive behavior, camouflage, or the habitats they inhabit. They may be naturally secretive or live in environments that are challenging for humans to access, making them less visible and difficult to study [6]. This is reflected in the low number of publications of A. basleri, even less, about their microbiome.

Therefore, any of the options that would allow us to increase the number of samples has its limitations and will generate variations in the results. However, we think studies based on small samples can contribute to our scientific understanding, and point to examples above. We are not claiming to have fully describe the microbiomes of these two species, but instead present a new approach to improve our understanding of wildlife without ethical or welfare concerns. 

Our study pioneers the study of wildlife microbiome using opportunistic sampling from roadkill in elusive and poor studied species. To our knowledge, this is also the first report of microbiota for A. bassleri or C. ani. And there are in fact, no report for related vertebrate species. As mentioned above and in the manuscript, first reports based on small samples sizes and without comparisons to live organisms are published and contribute to our understanding of nature (lines 327 – 329, applying tracking changes). This, together with the difficulty of collecting specimens in their natural habitat, make our results valuable to provide a baseline for these two species (lines 61-64, applying tracking changes). 

In addition to being a first report of microbiota in these two species, due to the characteristics of the samples we were also able to provide preliminary results to the question of whether the microbial community changed during the decomposition process of the animal (lines 67-70, applying tracking changes). 

The reviewer states that the “sample size is too small to eliminate any biases this limits the works ability to generalize the findings”. However, we do not claim (and we have rewritten parts of the text to ensure that is clear) to provide a comprehensive characterization of the microbiome in these two vertebrate species. Instead, our goal is to propose a new, non invasive method, showing it can offer insight into the microbiota of wild and difficult to sample species, and that initial decomposition (in more intact specimens as those selected in our study) does not result in a distinct microbiome.

The limitations regarding the sample size in this study have been addressed in various sections of the manuscript to ensure that all interpretations are made with this consideration in mind. Furthermore, we indicate that further research using similar approaches is needed to provide more information about the microbial communities in understudied wildlife species.

In summary, we agree with the comments of the Reviewer #1 regarding the limitations posed by a small sample size. However, we do not agree that should prevent the publication of this study because 1) there is no previous report from these two species (or related congeners) and the initial baseline can be valuable for further research; 2) although limited in number, samples were of suitable criteria and allowed us to describe microbiome; 3) highly biodiverse areas are at risk and mostly poorly studied, reducing this gap would be best served by promoting non invasive approaches, even if they come with some limitations, and also ensuring expectations from submitted work do not preclude work from the global south where resources and funding is more limited. 

We have improved the writing of the article to best clarify these points and have also expanded the description of the barcoding methods.

1. Coba-Males MA, Medrano-Vizcaíno P, Enríquez S, Brito-Zapat D, Martin-Solan S, Ocaña-Mayorga S, et al. From roads to biobanks: Roadkill animals as a valuable source of genetic data. PLoS One. 2023;18. doi:10.1371/journal.pone.0290836

2. Zhu J, Yu X, Zhang Q, Li Y, Tan S, Li D, et al. Cetaceans and microplastics: First report of microplastic ingestion by a coastal delphinid, Sousa chinensis. Science of the Total Environment. 2019;659: 649–654. doi:10.1016/j.scitotenv.2018.12.389

3. Fry B, Carter JF. Stable carbon isotope diagnostics of mammalian metabolism, a high-resolution isotomics approach using amino acid carboxyl groups. PLoS One. 2019;14. doi:10.1371/journal.pone.0224297

4. Senghor B, Sokhna C, Ruimy R, Lagier JC. Gut microbiota diversity according to dietary habits and geographical provenance. Human Microbiome Journal. Elsevier Ltd; 2018. pp. 1–9. doi:10.1016/j.humic.2018.01.001

5. Shanahan F, Ghosh TS, O’Toole PW. Human microbiome variance is underestimated. Current Opinion in Microbiology. Elsevier Ltd; 2023. doi:10.1016/j.mib.2023.102288

6. Franks DW, Noble J. Warning signals and predator-prey coevolution. Proceedings of the Royal Society B: Biological Sciences. 2004;271: 1859–1865. doi:10.1098/rspb.2004.2795

Reviewer #2: The manuscript "Gut Bacterial Communities in Roadkill Animals: A Pioneering Study of Two Species in the Amazon region in Ecuador" reports the gut microbes of Amphisbaena bassleri and Crotophaga ani. These two are wild animal species. Amphisbaena bassleri is a lizard whereas Crotophaga ani is a bird. There are no previous reports on the gut microbiome of these species as it is very difficult to collect samples from them. The authors have used the roadkill carcass of these animals to collect the samples. This is a very good strategy as the samples are collected without harming the wildlife. The authors performed NGS on the hypervariable V3-V4 region from the 16S rRNA. NGS of the hypervariable V3-V4 region from the 16S rRNA gene is a common method for studying bacterial diversity in microbial communities. The authors identified microbial phyla in both samples including Firmicutes, Bacteroidetes, and Proteobacteria for A. bassleri, and Firmicutes and Actinobacteria for C. ani. They also discuss the identified gut microbiota from Amphisbaena bassleri in light of the known literature from other reptiles. Also got microbiota from Crotophaga ani was discussed in reference to findings from other birds. Overall the authors clearly discuss their findings along with limitations of the study.

Answer to Reviewer #2

We would also like to sincerely thank Reviewer #2 for their time and comments on our work

---

## [Editor Report · Decision Letter 1]

22 Oct 2024

Gut Bacterial Communities in Roadkill Animals: A Pioneering Study of Two Species in the Amazon region in Ecuador

PONE-D-24-20575R1

Dear Dr. Poveda,

We’re pleased to inform you that your manuscript has been judged scientifically suitable for publication and will be formally accepted for publication once it meets all outstanding technical requirements.

Kind regards,

Sayed Haidar Abbas Raza

Academic Editor

PLOS ONE
---

## [Editor Report · Acceptance letter]

16 Dec 2024

PONE-D-24-20575R1 

PLOS ONE

Dear Dr. Poveda, 

I'm pleased to inform you that your manuscript has been deemed suitable for publication in PLOS ONE. Congratulations! Your manuscript is now being handed over to our production team.

Kind regards, 

on behalf of

Dr. Sayed Haidar Abbas Raza 

Academic Editor

PLOS ONE